# Nursing Students’ Perceptions of Clinical Debriefing TALK©: A Qualitative Case Study

**DOI:** 10.3390/nursrep15060194

**Published:** 2025-05-30

**Authors:** Belén González-Tejerina, Jorge Pérez-Corrales, Domingo Palacios-Ceña, Jose Abad-Valle, Paloma Rodríguez Gómez, Beatriz González-Toledo, Eva García-Carpintero Blas, Marta Garrigues-Ramón

**Affiliations:** 1Escuela Internacional de Doctorado, Research Line in Physical Therapy, Occupational Therapy, Rehabilitation and Physical Medicine, Universidad Rey Juan Carlos, Avenida Atenas, s/n, 28922 Alcorcón, Spain; b.gonzalezt.2018@alumnos.urjc.es; 2Research Group of Humanities and Qualitative Research in Health Science (Hum&QRinHS), Department of Physical Therapy, Occupational Therapy, Rehabilitation and Physical Medicine, Universidad Rey Juan Carlos, Avenida Atenas, s/n, 28922 Alcorcón, Spain; domingo.palacios@urjc.es; 3Fundación Jiménez Díaz School of Nursing, Autonomous University of Madrid, Av. de los Reyes Católicos, 2, 28040 Madrid, Spain; jose.abad@quironsalud.es (J.A.-V.); prodriguezg@quironsalud.es (P.R.G.); beatriz.gonzalezt@fjd.es (B.G.-T.); marta.garrigues@quironsalud.es (M.G.-R.); 4Health Research Institute-Fundación Jiménez Díaz University Hospital, Autonomous University of Madrid (IIS-FJD, UAM), Av. de los Reyes Católicos, 2, 28040 Madrid, Spain; 5Grupo NBC, Departamento de Salud, Facultad de Ciencias de la Vida y de la Naturaleza, Universidad Nebrija, 28248 Madrid, Spain; egarcibl@nebrija.es

**Keywords:** clinical debriefing, nursing students, reflective practice, qualitative research

## Abstract

**Background/Objectives**: Clinical debriefing is a learning tool that promotes reflection after critical incidents, improving patient safety and professional performance. TALK© is a debriefing technique designed to facilitate structured team self-reflection after any learning event in clinical settings. The aim of this study is to explore the experiences of fourth-year nursing students in clinical internships with clinical debriefing guided by the TALK© tool. **Methods**: A qualitative case study was conducted. Twenty-seven participants were recruited using purposeful sampling. The sample consisted of nursing students. Data were collected through in-depth interviews, focus groups, personal writings, and researcher field notes. An inductive thematic analysis process was applied. The data analysis was performed using ATLAS.ti 23 software, which facilitated the identification and organization of key themes and patterns within the qualitative data. **Conclusions**: Participants perceived TALK©-guided clinical debriefing as a valuable practice. Key factors influencing their experience included the reflexive process, the debriefing approach and technique, the timing and context, as well as its emotional sphere.

## 1. Introduction

The increasing complexity of patient conditions and the evolving medical landscape demand high-quality healthcare services to ensure optimal patient outcomes [1].

Debriefing is a learning method based on reflection after a real or simulated clinical event that contributes to clinical excellence and patient safety. Clinical debriefing involves a structured conversation conducted by a facilitator, whose objective is to analyze a real clinical situation in order to reinforce appropriate behaviors and improve those susceptible to change, promoting better performance and patient safety [2]. Structured debriefing after critical incidents is recognized as a pivotal component in enhancing patient safety and fostering a culture of continuous learning within healthcare systems. The World Health Organization’s Global Patient Safety Action Plan 2021–2030 emphasizes the implementation of such practices to mitigate harm and improve care quality [3].

Further, debriefing is highlighted as an essential strategy for clinical learning and team performance enhancement [4], and for linking quality improvement with the well-being of healthcare professionals [5].

Clinical debriefing fosters team communication, allowing team members to reflect on their experiences, support each other, and agree on points for improvement [6]. There are different approaches and techniques for clinical debriefing. The judicious debriefing approach focuses on actively listening to learners and respecting all responses [7,8].

The TALK© tool is designed to improve patient safety through short, structured debriefs, focusing on positive and constructive reflection after clinical events [6].

### 1.1. Patient Safety Education Among Nursing Students

Patient safety is a global priority in healthcare systems, and the World Health Organization has urged the early integration of related competencies in professional training [3].

Evidence shows that structured educational programs on patient safety significantly enhance nursing students’ knowledge, attitudes, and skills, fostering a preventive culture from the early stages of education [9].

Furthermore, the need for innovative pedagogical approaches is widely acknowledged for developing critical thinking and safe decision-making in real-life scenarios [10]. One of the most notable is the gap between theoretical instruction and its practical application during clinical placements. Farokhzadian, Torkaman, and Sabzi [9] identified discrepancies between students’ self-perceived competencies in classroom versus clinical settings, highlighting the need for more experiential and context-driven teaching strategies. In this regard, active learning methodologies—such as simulation, case analysis or clinical debriefing tools—have proven highly effective in reinforcing meaningful learning and encouraging critical reflection [10,11].

### 1.2. TALK© Debriefing

The TALK© tool was developed with the aim of improving patient safety and fostering a culture that promotes conversation and dialog by guiding teams to conduct structured, solution-focused debriefs after everyday events that can serve as learning opportunities.

Conducting a clinical debrief using the TALK© tool involves gathering the team after a clinical event to have a conversation that fosters learning in a positive and constructive manner. The debrief should last no longer than 10 min and can be conducted immediately after the critical situation or at a more suitable time, depending on the circumstances, such as when team members are exposed to new situations. The purpose of clinical debriefing using the TALK© structure is to analyze the event, reflecting on the behaviors that led to success and those that require improvement, with the team taking responsibility for ensuring that actions needing improvement are addressed in future events [6].

The TALK© tool has been successfully used and translated into other languages, allowing its implementation. There are currently initiatives in 18 countries across five continents [12].

Evidence indicates that clinical debriefing is particularly beneficial for professionals in critical units such as Intensive Care Units (ICU), operating rooms and emergency rooms, and improving learning and professional performance [13]. Despite the widespread application of debriefing techniques in simulated clinical settings, there is a notable lack of research on their use in real-world clinical environments with undergraduate nursing students. This study addresses this gap by examining how structured debriefing impacts students’ learning and emotional management during critical incidents in clinical practice.

Based on this, clinical debriefing guided by the TALK© tool, using a good judgment approach, was implemented with fourth-year nursing students.

The aim of this study was to address a gap in the existing literature by examining how structured debriefing, specifically through the TALK© model, influences nursing students’ learning and emotional management during critical incidents in clinical practice.

## 2. Materials and Methods

The research question that guided this qualitative case study is the following:

How does structured debriefing using the TALK© model influence nursing students’ learning and emotional management during critical incidents in clinical practice?

The research questions shaped the methodology, ensuring that interviews, focus groups, and personal writings captured diverse perspectives. Each method was aligned with the central questions regarding students’ perceptions of debriefing and its impact on their learning and emotional well-being.

### 2.1. Design

A case study approach was selected to provide an in-depth exploration of nursing students’ perceptions of clinical debriefing using the TALK© tool. This method aligns with Yin [14] and Creswell and Poth [15], as it allows for the examination of complex, contextualized phenomena in real-world settings. The qualitative case study design facilitated a nuanced understanding of student experiences and their reflections, essential for addressing the research questions.

This qualitative case study explores a clinical debriefing program in its entirety, treating it as a single unit of analysis [16]. Using a descriptive approach [17], the study presents a detailed account of the phenomenon without being anchored to a predefined theoretical framework.

This study examines the perceptions of fourth-year nursing students regarding clinical debriefing, following their supervised clinical practice in specialized care settings where they encountered critical situations. It was conducted in accordance with the Consolidated Criteria for Reporting Qualitative Research (COREQ) and the Standards for Reporting Qualitative Research (SRQR) [18,19].

In addition, Guba and Lincoln’s criteria for establishing the reliability of the data were considered by examining issues related to the credibility, dependability, confirmability, and transferability of the data [20].

### 2.2. Research Team

The research team consisted of seven nurses (BG-T, EG-CB, BG-T, MG-R, JA-V, PR-G, and DP-C) and one occupational therapist (JP-C). Seven of them (BG-T, EG-CB, DP-C, JP-C, JA-V, MG-R, and PR-G) had experience in qualitative research. Two of them (BG-T and EG-CB) were instructors in simulation and debriefing. One of the researchers (EG-CB) facilitated the debriefing sessions and was not involved in data collection through interviews or focus groups, nor in the data analysis process. The remaining researchers were not present during the debriefing sessions to avoid influencing the results obtained.

One of the researchers (BG-T) served as the academic tutor for some participants and had a pre-existing relationship with them. This prior connection could introduce bias; however, it may also be viewed as a strength, as it likely fostered a climate of trust and openness during data collection.

### 2.3. Setting

This study was conducted at the Fundación Jiménez Díaz School of Nursing, affiliated with the Autonomous University of Madrid, where simulation-based education (SBE) is an integral component of the teaching methodology [21]. Regarding the approach, at the Fundación Jiménez Díaz School of Nursing, debriefing is used with a good judgment approach.

The nursing degree program includes a substantial number of supervised clinical practice hours, particularly from the second to the fourth year. Over the course of their training, students complete approximately 2300 h of clinical practice [22], during which they are frequently exposed to complex care situations. From the third year onward, students begin rotations in specialized units—such as medical and surgical ICUs, emergency departments, and operating rooms—where they may encounter critical incidents.

### 2.4. Participants

The study population consisted of fourth-year students from the Fundación Jiménez Díaz School of Nursing. Inclusion criteria included having completed supervised clinical practice in specialized units (ICU, emergency department, or operating room) during the academic year; having experienced critical incidents during these internships; having participated in clinical debriefing sessions within the aforementioned units; and being Spanish-speaking.

Convenience sampling was used, allowing the selection of participants who met the inclusion criteria and were accessible to the researchers due to their proximity to the study setting. Ultimately, 27 participants were recruited through face-to-face engagement.

### 2.5. Data Collection and Analysis

Data collection was carried out in three phases. The first phase took place in March 2023, and involved unstructured in-depth interviews with five participants to identify the most relevant topics for the elaboration of the semi-structured interview guide used in the second phase. The second phase of data collection took place between April and May 2023 and involved three focus groups, conducted according to the service where the students were completing their internships and participated in clinical debriefing. These focus groups aimed to explore students’ perceptions of clinical debriefing, using a semi-structured script (Table 1) previously prepared based on interviews with the first five participants, a review of the literature on the subject and the researchers’ expertise in the field.

Secondly, 26 semi-structured individual interviews were conducted with the participants of the three focus groups, adding questions to the script to gather more information. Lastly, 20 personal documents were collected to capture the experience of the participants. In addition, data collection was supplemented with field notes from the researcher (BG-T). Both the focus groups and the semi-structured individual interviews were conducted by one of the researchers (BG-T). This researcher was not present during the debriefing process. The focus groups were audio- and video-recorded; therefore, the presence of an observer was deemed unnecessary, and only one moderator (BG-T) was present. The interviews were recorded using audio only. Subsequently, the transcription of the three focus groups (Table 2) and 26 individual semi-structured interviews were carried out. The focus groups had an average duration of 99 min. The semi-structured individual interviews lasted on average 60 min, with the longest lasting 130 min and the shortest 37 min.

A full verbatim transcription of each of the focus groups, interviews, reflective diaries, and researcher’s field notes was completed. The field notes were used in the analysis as they provided greater perspective and added depth to the data and/or verified the data obtained in the interviews (triangulation). All participants were offered the opportunity to review the audio or written records as well as the subsequent analysis to confirm the researchers’ interpretation of their experience. No additional comments were made by any of the participants.

The inductive thematic analysis process proposed by Braun and Clarke [23] was applied. This is an iterative process of moving back and forth through the entire dataset, where the interviews were fragmented into narrative segments to which different codes of meaning were assigned. These codes were grouped into different thematic levels (categories, subthemes, and main themes) with the help of tables and thematic maps, and were described, reflecting the participants’ experiences and accompanied by narratives that ensured the traceability of the results. The inductive thematic analysis followed Braun and Clarke’s six-phase framework [24]: (1) Familiarization with the data: Researchers immersed themselves in the dataset through repeated readings, taking notes and identifying initial patterns; (2) generating initial codes: narrative segments were coded to capture relevant features, while contextual elements not linked to specific codes were retained to aid interpretation; (3) searching for themes: codes were organized into categories and overarching themes through a collaborative process. Themes were built across multiple levels and refined using data tables for cross-checking; (4) reviewing themes: themes were examined and revised for internal consistency and relevance, discarding, merging, or splitting them as needed. The full dataset was re-read to ensure coherence and completeness; (5) defining and naming themes: the core meaning of each theme was clarified, and excerpts were reorganized to support clear, distinct thematic descriptions; (6) producing the report: a coherent narrative was developed, supported by illustrative excerpts and connected to the research question and the existing literature.

For the determination of the initial sample size, the researchers did not adopt a strict interpretation of the saturation criterion, commonly defined as the point at which no new themes or codes emerge, as discussed by Braun and Clarke [24]. Instead, saturation was approached as a flexible and ongoing process, prioritizing the depth, richness, and coherence of the data in relation to the study objectives and the analytical framework used. It is important to note that saturation does not depend solely on the number of interviews conducted, but rather on the quality, depth, and density of the data obtained, as well as the type of analysis performed. In qualitative designs, saturation can typically be achieved with samples ranging from 12 to 30 participants, depending on the context and the homogeneity of the group, as noted by Guest et al. [25] and Turner-Bowker et al. [26]. In this study, an initial sample size of approximately 30 students was estimated, based on the upper threshold identified by Turner-Bowker et al. [26], who found that 99.3% of key concepts emerge within the first 30 interviews. Nevertheless, data collection concluded with participant number 27, at which point no new relevant analytical categories were identified, and thematic saturation was considered to have been reached.

The initial analysis was conducted by one of the researchers (BG-T) using ATLAS.ti software. Subsequently, a separate analysis was performed by another researcher (JA-V) to enable researcher triangulation. These researchers were not in contact with each other or with the participants during the debriefing process. After the initial independent coding conducted by researchers BG-T and JA-V, a comparative and interpretative phase was undertaken. In a series of review meetings, the researchers examined and discussed the codes, themes, and subthemes identified by each of them. Through reflexive dialog, they explored points of convergence and divergence in their interpretations, paying close attention to both the content and contextual meaning of the data segments. Finally, the entire analysis process was reviewed by a third researcher (JP-C), agreeing on the themes, subthemes, and categories by inter-researcher consensus. Researcher JP-C acted as a methodological facilitator during this stage, overseeing the internal coherence of the thematic framework and supporting the decision-making process grounded in the study’s aims, the consistency with the adopted qualitative approach, and the interpretative depth of the material. Consensus was not understood as the lack of disagreement, but rather as the co-construction of an analytical framework that integrated diverse perspectives, honored the complexity of the data, and enhanced the overall credibility of the analysis [27,28].

### 2.6. Ethical Considerations

Permission was obtained from the Ethics Committee of the Fundación Jiménez Díaz Hospital to carry out the present study under code PIC216-22_FJD. All participants were given a participant information sheet, informed consent, and consent to video and audio recording. The ethical principles promulgated by the Declaration of Helsinki [29] were complied with.

## 3. Results

Sociodemographic data were collected from the participants, such as age and sex (Table 3). Participants P1, P3, and P4 took part in the pilot study of the first phase; therefore, they are not reflected in the table. Participant P2 participated in both the pilot study and the final data collection phase, as their clinical rotation month in the operating room coincided with the study period. During the pilot phase, P2 was in the ICU.

Six clinical debriefing sessions took place, consisting of students who were completing their clinical rotations in the ICU, ER, and/or OR. Additionally, the focus groups were composed of twenty-seven out of the thirty students who had participated in the clinical debriefing sessions related to their rotation unit, resulting in the formation of the ICU, ER, and OR focus groups. Three students from the OR debriefing did not participate in the study, and one student did not take part in the focus group but did participate in the semi-structured interview. Conversely, another participant from the OR debriefing took part in the OR focus group but not in the individual interview.

After analyzing the data from three focus groups with a total of 27 participants, 26 individual semi-structured interviews, and 20 personal documents of the participants’ experience, 332 codes were obtained and grouped into 24 categories. Three subthemes were established which were included in the theme process and structure of clinical debriefing and two subthemes were included in the theme The Emotional Sphere (Figure 1). A thematic table summarizing the structure of the analysis has been included as Appendix A, to provide a clear and transparent overview of the analytic process.

### 3.1. Process and Structure of Clinical Debriefing

The theme process and structure of clinical debriefing addresses in three subthemes, reflexivity, approach and technique, and moment and context, how participants perceived the process and structure of clinical debriefing and the factors they considered influential in it.

#### 3.1.1. Reflexivity

For participants, reflexivity in clinical debriefing is rooted in a structured process that enhances and formalizes the natural reflection triggered by meaningful clinical experiences—whether positive or negative—ultimately contributing to improved performance. Guided approaches, such as the TALK© tool, support in-depth analysis and the identification of areas for improvement. However, participants also emphasized the value of everyday reflection in clinical practice, regardless of the specific tool used.


*P19: if you experience a situation and it is a situation that impacts you in some way, good or bad, innately you are already going to reflect on that situation [...]. Now, doing a debriefing dedicated to that specifically is going to help you more than doing it by yourself in a natural way, it is giving you an objective, which is to look for points of improvement and to analyze the situation.*


Participants highlighted that setting debriefing objectives boosted motivation and openness to reflect. Facilitators’ questions deepened reflection and increased awareness of actions. Retrospective reflection in clinical debriefing enabled better analysis by reducing emotional immediacy. They found clinical debriefing more profound due to real experiences, compared to the technical, less reflective nature of educational debriefing in simulations. Greater knowledge and experience fostered more active participation, but equitable and collaborative participation was emphasized to ensure patient care quality.


*P26: if you are the one who decides it is because you want to draw conclusions from these things or solutions and learn. I think you are more open to reflect.*



*P5: It is as if you have experienced something at one moment and you are going to reflect on it at a different moment. You are going to better integrate that knowledge and the memories you have.*



*P26: With limited knowledge, you can’t engage in deeper reflection.*



*P6: In a debriefing, it’s not the person who knows the most who should speak the most—especially when the goal is to learn how to manage patient care effectively in an interdisciplinary team. Everyone needs to participate equally.*


Participants emphasized the importance of learning from clinical errors through clinical debriefing for continuous improvement. In addition, participants contemplated the possibility that error management could be complicated due to the fear of judgment. However, for them, the trusting environment that was generated facilitated open and productive reflection. Participants reflected on a wide range of critical incidents to improve patient safety. Thus, participants described how becoming aware of their own mistakes served as a catalyst for critical thinking. This active reflection, emerging during clinical practice, enabled them to question automatic habits and promoted more deliberate decision-making.


*P20: We discussed a case in the operating room. There was laparoscopic digestive surgery, and they ended up sectioning the hepatic artery, and then there was massive bleeding, and we analyzed what we thought we could have done or what could have been done to avoid it or how the thing flowed when it happened.*



*P13: You start developing more critical thinking when you become aware of the mistakes you make. When you continue doing clinical practice, you stop and ask yourself… why am I doing this, and why not something better? You train your mind not to do things automatically.*


For the participants, the protagonist of a critical incident tended to have a deeper reflective process due to first-hand experience. However, the willingness of such a protagonist to share experiences, they thought, might be limited by intense emotions, especially if the situation was difficult or with negative consequences for the patient. For some participants, this could limit the capacity and depth of reflexivity. In this sense, participants valued learning through multiple perspectives, beyond that of the main protagonist. This allowed them to broaden their understanding of clinical situations, recognizing that shared reflections contributed to more comprehensive and less self-centered interpretations. Engaging with others’ viewpoints helped them reframe their own experiences and find more practical, less emotionally charged solutions.


*P6: What touches you up close is engraved in your heart, isn’t it? That is so. Others will also learn something, but it won’t be so engraved, it won’t remind you of that patient, it may remind you of what we talked about in the debriefing.*



*P5: If you share something that happened to you, you might feel more exposed—maybe, I don’t know, it depends on the situation—but possibly more insecure depending on what actually happened.*



*P17: For me, it was a moment that provided space to express if you have any other concerns, and to know that you can count on others who are in the same situation as you. And to be able to solve that problem from different perspectives. And in that way, you can find a practical solution.*


#### 3.1.2. Approach and Technique

Participants perceived the facilitators—external and neutral figures—as effectively guiding the debriefing using the TALK© tool with a good judgment approach and a natural style. While the tool provided a structured framework, the facilitators’ flexible application of it helped ensure that participants did not feel inhibited during the process. The facilitators’ impartiality was a highly valued aspect. As they had not witnessed the events firsthand, they maintained a neutral, observer role, which, according to participants, fostered a safe environment where students felt comfortable sharing their experiences openly and without self-censorship. All this focused the debriefing on learning by providing structure and freedom of expression.


*P6: It is a good way to get to the end of the debriefing, which is to look for that solution. It allows you to have a much more holistic, much more general view of the case, and I think that also allows you to analyze it better, reflect better and draw better conclusions. So, I think it is a very good tool.*



*P10: I think they were guiding us with the TALK tool, because otherwise... we would have gone off track. But it wasn’t super rigid—like, if we were on the ‘L’ step, we weren’t forbidden from going back.*



*P19: I think it has to be someone external, neutral, and who handles the tool well, for it to be effective.*


Having shared internships improved the comfort and quality of the debriefing, creating an environment of mutual support and understanding. Previous experience with the TALK© tool also increased the comfort of some participants who had previously participated in another clinical debriefing.


*P5: You are with them every afternoon. So, in the end, we have that feeling, we are comfortable there. [...] As I already had the previous experience of the ICU... better, you get to know the tool better and you follow it more, you are more comfortable.*


The participants’ choice of debriefing objectives increased their involvement and commitment, making them feel important and listened to. The imposition of objectives on them could reduce learning. However, participants also acknowledged that, although addressing certain topics might feel uncomfortable, there are situations in which facilitators may need to take the initiative for the benefit of the group and collective learning.


*P5: If we do not choose the topics to talk about and one is imposed on us, maybe we are not so willing to comment on that case or maybe we have not experienced anything similar.*



*P8: Sometimes you don’t bring up the topic yourself… for whatever reason, and if the facilitators know what happened and that it’s useful to talk about it, it might be important.*


Participants noted that the facilitators’ questions during clinical debriefing were designed to deepen reflection. These questions helped students become more aware of their actions, assess both the positive and negative aspects of situations, and boost their self-esteem by recognizing their own achievements. The facilitators’ questions were formulated with curiosity and respect for the participants and helped them to deepen their reflection, break the ice, and generate closeness. The familiarity with the facilitators facilitated the response to their questions. Familiarity also allowed students to feel more comfortable and open to sharing their thoughts and feelings, even when facing potentially uncomfortable questions.


*P22: They were questions to make you think about why, and not to judge you or question you, but to reflect.*



*P26: In the end, since we knew them, it made it easier—so even if they asked a question that might have made us uncomfortable, because we already knew them, it wasn’t hard to answer, you didn’t second-guess your response—you just answered.*


#### 3.1.3. Moment and Context

Participants offered different perspectives on how the time elapsed between the event and the debriefing influenced their ability to process what had happened, manage their emotions, and learn from the situation in a meaningful way. The students remarked on the importance of separating experience from reflection to better integrate knowledge and memories. However, performing clinical debriefing as soon as possible could be beneficial to prevent future errors quickly.


*P6: Not in the heat of the moment. Especially if it has to be something negative. I think that after... after a week or so, or even a few days, that’s when it’s better. [...] I mean, out of the storm everything is much clearer.*



*P7: the sooner you talk about it, the sooner other people can also realize that I can also make a mistake in that and you can also prevent other people’s mistakes.*


The participants noted that, although the structure was consistent, the usefulness of clinical debriefing varied according to the setting and the specific characteristics of each hospital unit. The applicability of debriefing-derived solutions was more relevant in the ICU. Furthermore, in the ICU, debriefing tended to be more informal due to a closer relationship between colleagues, whereas in the operating room it was more formal and distant. The students acknowledged that the first experiences in clinical practice, both in emotional and learning terms, were highly relevant moments, which they perceived as useful to analyze in the debriefing.


*P21: there are differences because, well, each service is different. So, well, debriefing? I think that the dynamics of debriefing would not really change. It would simply change because of the content or errors that may occur, which may be more serious.*



*P19: In the end, I think that the ICU is an internship where students have a lot of relationship with the rest of the classmates, so you develop more confidence. In other internships where you may not have as much connection, debriefing can be difficult, as in the operating room.*



*P5: But the fact that it is a first time is like you have never experienced anything like it before. So that marks you, you are not used to it, and it impacts you. And you take that with you. And it’s important, it’s like you learn more from it.*


Participants also compared clinical debriefing with informal conversations during breaks. Participants referred to “break-time conversations” as informal reflective moments comparable to clinical debriefing. During breaks in their clinical placements, nursing students found a relaxed space to share experiences and emotionally unwind with peers. However, they noted that unlike clinical debriefing, these conversations were less structured and lacked deep reflective focus. Students perceived them more as opportunities for emotional release and disconnection than as formal learning processes. Unlike debriefings, these moments rarely involved structured reflection; instead, students tended to recount events without actively seeking solutions or lessons.


*P6: Clinical debriefing has a real objective for the well-being of the patient and to get something out of it. What we share during snack breaks is often not a reflective process, it is an absolute avoidance process.*


Implementing clinical debriefings in each internship was seen as beneficial for the participants. Initiating debriefing from the first year helped students to acquire the reflective habit. However, the reflection was deeper in the more advanced years. In the last years of the degree, clinical debriefing became more relevant and necessary as greater responsibilities were assumed, and the work environment was closer.


*P21: Let’s see, it could be done in the second... in the third and fourth year to see the evolution. I think it would never hurt. But now that’s it, you’re on the edge, you’re on the edge of student life and what I’ve learned in it, and the abyss of... “hello, let’s see what comes to me.”*


### 3.2. The Emotional Sphere

The theme The Emotional Sphere encompasses two subthemes, The Emotion of Experience and Emotional Management and Support. It captures the emotions and feelings that participants experienced during the clinical debriefing, highlighting how emotional bonding and the management of judgment and blame significantly influenced their learning, reflection, and emotional well-being.

#### 3.2.1. The Emotion of the Experience

The study participants highlighted the crucial importance of emotional bonding during the clinical debriefing process. Intense emotions experienced during real situations had a significant impact on learning and reflection. In contrast, in a simulated debriefing, emotions were more superficial, limiting the ability to reflect. Several participants emphasized that clinical debriefing served as a bridge that brought them closer to the lived experience and helped them retain knowledge in the long term. They described how, in real-life situations, emotions were more intense, which facilitated reflection and led to a deeper understanding of the event. Moreover, they reported feeling more motivated and interested in actively participating in clinical debriefing when the situations involved real patients they had interacted with personally. Real situations fostered greater engagement during debriefing. However, cases that involved errors carried significant emotional weight for some participants due to feelings of responsibility, which sometimes hindered open discussion. In contrast, educational debriefings, while recognized as useful, were perceived as less impactful and elicited lower levels of involvement, given their lack of real-world consequences.


*P5: I feel that learning is more visible, i.e., you assimilate it more if you relate it to a feeling that it has produced in you, it is more marked.*



*P7: if something goes wrong in a simulation it doesn’t affect me, because I say, well, I did it wrong, but nobody got hurt. [...] In the clinical debriefing, if we comment on the fact that I have inadvertently made a medication error, then I know that I will get more involved.*


#### 3.2.2. Emotional Management and Support

Participants identified balancing learning and guilt as key in clinical debriefing. Participants thought that the feeling of guilt was intrinsic to the situation experienced but not to the debriefing, especially in cases of medication errors, and unavoidable. Regarding guilt, the imposition of objectives and having their care coach as the facilitator could have heightened feelings of judgment, hindering honest expression.


*P6: I felt a bit like saying, it wasn’t that hard either [...] So, on the one hand, learning, and on the other hand, it was like, you could have done better. Like feeling, it’s not like feeling dumb, but a little bit of guilt.*



*P19: Yes, there was some issue that could... could lead to that because we talked about medication errors and things like that. But other than that burden of guilt that already comes with the issue innately, no, I didn’t feel judged or anything like that.*



*P20: I think if our care coach is the one leading the debriefing, you hold back a little… you worry more about being judged, especially if you didn’t handle something well.*


Peer support was crucial in minimizing feelings of guilt and providing mutual support. Trust and confidence increased when they felt they could talk without being judged, fostering greater openness and communication. The clinical debriefing was perceived as valuable for the expression of emotions and feelings. This helped participants to normalize them, understanding that they were not alone in their feelings. Participants emphasized the importance of considering the emotional sphere within the facts for a complete understanding of the situations experienced.


*P8: I think that with the group I had it was minimized, that is, if P6 could feel guilty, then the rest of the colleagues made him feel that he was not guilty.*



*P15: It is very hard to open up and say what we feel, and thanks to this you can say, well, look, I was so scared that I ran away. It makes it easier to talk. You don’t always have the opportunity or the space to express your feelings.*



*P10: Well, a little bit of what my classmates have said about, well that, by talking about it and so on, knowing that it could not only have happened to you, and also feeling supported and listened to a little bit.*


## 4. Discussion

### 4.1. Process and Structure of Clinical Debriefing: Reflexivity, Approach and Technique, and Moment and Context

The concept of reflection has long been a subject of study [30,31,32,33] and continues to be relevant today [34,35]. The participants in our study stressed the importance of constant reflection to improve their practices, in line with Healy and Murphy [36], Barchard [34], and Galligan et al. [37], who emphasize the relevance of structured reflection for continuous professional development. In particular, retrospective analysis of incidents, conducted at a separate point in time from the event, allowed for deeper reflection by lessening the immediate emotional burden on our participants, as described by Schön [38] in his concept of reflection on action. Ekebergh [39] and Finlay [40] proposed that retrospective reflection, informed by context, allows participants to reconstruct details of the event to gain new insights and improve practice [41]. More recently, this perspective has been reinforced by the Healthcare Simulation Standards of Best Practice™ [42], which emphasize that the ultimate goal of the debriefing process is to promote reflective thinking and cognitive reframing. This process encourages learners to integrate new insights with prior knowledge, thereby strengthening clinical reasoning and professional development.

There is debate as to whether reflective practice should be applied during the action or after the action, with reflection on the action being the most common in nursing practice [43]. The time between the incident and debriefing facilitated more balanced and less emotional reflection, in line with Epp [44] and Twigg [45], who also indicated that temporal separation between experience and reflection enhances knowledge integration. However, some participants in the study by Galligan et al. [37] affirmed that elapsed time did not influence their experience in clinical debriefing. The depth of reflection was influenced by the level of knowledge, according to our participants, in which Galligan et al. [37] concurred. Other participants in our study emphasized the importance of fairness and horizontality regardless of knowledge level, which is consistent with best practices in debriefing that promote an environment of psychological safety and mutual support, as described by Healy and Murphy [36].

Our participants reflected on a wide range of critical incidents to improve patient safety, an aspect addressed in other studies [2,13]. Recent studies have examined the effectiveness of the TALK© tool. For example, Díaz-Navarro et al. [46] conducted a study on the implementation of the TALK© tool in operating theaters of a hospital in the United Kingdom, evaluating its effects on team behavior and continuous patient safety improvement. Their findings indicate that the introduction of TALK© significantly increased both the consideration and execution of debriefings among surgical teams, with initial consideration rates rising from 35.6% to a range between 60.3% and 97.4% following the intervention. The frequency of completed debriefings also rose, reaching 39% in the six-month follow-up. Moreover, action planning for continuous improvement increased from 17.6% to 70% within the first six months. The study concludes that the TALK© tool, due to its ease of use, low cost, and inclusive approach, is an effective strategy for promoting continuous improvement and patient safety in surgical settings.

Reflection without a formal framework reduces engagement and effectiveness [47], with a structure such as the good judgment approach and tools such as TALK© being important, the structure of which appears to align with others such as Kolb’s learning cycle and Gibbs’ reflective cycle [48,49,50]. The organic and spontaneous implementation of the TALK© tool along with the questions asked by the facilitators were valued by our participants, facilitating an environment of trust and participation, as described by Sandars [51]. This method aligns with Rudolph et al. [52] and Fey et al. [53], who highlight the importance of combining rigorous feedback with genuine consultation for effective debriefing.

Familiarity with the TALK© tool improved the experience of our participants, who experienced it as more fluid and comfortable, as recommended by Diaz-Navarro et al. [6].

All these findings are consistent with Edmondson and Lei’s [54] model, which postulates that psychological safety is essential for team learning. Recent studies continue to affirm the critical role of psychological safety in fostering effective team learning. For instance, Clausen et al. [55] found that psychological safety significantly enhances team creativity and academic well-being among first-year university students engaged. Similarly, Jin and Peng [56] demonstrated that team psychological safety positively influences innovative performance, with communication behavior serving as a mediating factor.

Unit-specific characteristics influenced the perception of clinical debriefing. In the ICU, debriefing was more informal due to the closeness between peers, whereas in the operating room it was more formal and distant. Tailoring debriefing to the specific context of each clinical setting is crucial to maximize its effectiveness [36]. The implementation of clinical debriefing at different stages of nursing education was perceived as beneficial, helping to acquire the habit of reflection from the early years of career and being deeper reflection the last years of career, agreeing with Roca et al. [57]. Informal conversations during breaks were also valued by our participants. This informal approach, while valuable for emotional well-being, does not provide the same level of structured learning and deep analysis as formal debriefing [44], as referenced by our participants. However, some studies show a preference for informal meetings over structured debriefing [58].

### 4.2. The Emotional Sphere: The Emotion of Experience and Emotional Management and Support

Our participants emphasized the importance of emotional bonding during clinical debriefing, a finding consistent with previous studies. McCloughen et al. [59] noted that intense emotions experienced in real situations can significantly impact learning and reflection, which is consistent with our participants’ observations. The emotional intensity of real clinical experiences, compared to simulated ones, resulted in greater engagement and deeper assimilation of learning. However, situations involving errors generated considerable emotional weight due to the perception of responsibility, which could hinder open participation in debriefing. This is aligned with the study by Andersen et al. [60] reporting that students come to discussions with the underlying intention of not speaking up due to the fear of being judged. Furthermore, educational debriefings, perceived with less seriousness due to the lack of real repercussions, resulted in less emotional involvement. Judgment and guilt management were key issues for our participants. Perceived judgmentalism, especially when the assistive coach was the facilitator, hindered participation and honesty. Peer support emerged as crucial in minimizing feelings of guilt and providing a mutually supportive environment. This finding is consistent with Arriaga et al. [13] where the importance of a safe and structured environment for students to reflect on their experiences and receive feedback is highlighted. Debriefing was perceived as a valuable tool for the expression of emotions and feelings, helping participants to normalize their emotional experiences and understand that they were not alone in their feelings, coinciding with Plowe [61]. This aspect also coincides with the study by Elarousy et al. [62] that highlights the need for a safe and supportive environment during debriefing to facilitate openness and communication among students. The support of colleagues emerged as crucial in minimizing feelings of guilt and providing a mutually supportive environment. This finding is consistent with the observations of Fey et al. [53], who highlighted the importance of peer support in facilitating emotional expression and deep reflection. Recent studies continue to reinforce this perspective. For example, Parmar et al. [63] found that a peer support program significantly enhanced the social and emotional well-being of postgraduate health students during the COVID-19 pandemic by reducing stress and fostering emotional resilience. Similarly, Abraham and Singaram [64] demonstrated that peer feedback, when conducted within a trusting environment, enhances receptivity and engagement among medical students, promoting effective self-regulation and reflective learning. These findings underscore the importance of establishing peer support networks that value shared experience, thus facilitating more meaningful and deeper reflective processes. Clinical debriefing is a relatively novel field of practice and research, even more so in undergraduate students. In recent years, multiple tools have been developed to conduct it [65].

Regarding practical implications, the findings of this study suggest that implementing structured debriefing tools such as TALK© in undergraduate nursing education may enhance students’ ability to process clinical experiences critically and manage emotional responses more effectively. Integrating TALK© into post-clinical reflection sessions could support the development of clinical reasoning, emotional resilience, and communication skills. Moreover, adopting a standardized debriefing framework may foster a culture of psychological safety and reflective practice from early stages of professional training. This could ultimately contribute to improved preparedness for clinical decision-making and greater patient safety outcomes. Educators should consider incorporating structured debriefing methods into nursing curricula and offering faculty training to ensure consistent, supportive implementation.

Although including only final-year students—due to their greater clinical exposure—was a deliberate methodological choice, it also represents a limitation, as the long-term impact of TALK© debriefing on learning and clinical practice remains unknown. Since fourth-year students are nearing graduation, future research could benefit from including third-year students to examine TALK©’s effects over a longer timeframe.

Another limitation is that, although none of the researchers involved in data collection and analysis were present during the debriefings themselves, one of the researchers (BG-T) served as the academic tutor for some participants and had a pre-existing relationship with them. This prior connection could introduce bias; however, it may also be viewed as a strength, as it likely fostered a climate of trust and openness during data collection.

A key strength of this study is its qualitative exploration of nursing students’ experiences with clinical debriefing, specifically the TALK© model, which has not previously been studied in undergraduate nursing education. This approach allowed for a rich understanding of student perceptions, adapting to emerging insights and capturing nuanced individual experiences in depth. Furthermore, the study provides an original perspective on the educational value of debriefing frameworks in undergraduate settings and offers practical insights that may inform the integration of structured debriefing tools into nursing curricula.

Building on the findings of this study, future research could involve similar studies with third-year nursing students to explore the longer-term educational impact of the TALK© debriefing model. Including students earlier in their training would allow for the observation of how structured debriefing influences learning trajectories and professional development over time. Additionally, given the strong emphasis placed by participants on the emotional dimension of the debriefing process and considering that the TALK© model does not explicitly incorporate a component dedicated to emotional processing, there is a compelling rationale to develop or adapt debriefing frameworks to address this gap. Designing a structured tool that intentionally integrates emotional reflection could enhance the depth and effectiveness of post-clinical debriefing, particularly in emotionally charged learning environments such as those encountered in nursing education.

## 5. Conclusions

Clinical debriefing, using tools like TALK©, is seen as a valuable practice for deep reflection, learning, growth and promoting patient safety. Key factors include reflexivity, technique, timing, and context. Regular implementation during internships benefits students by providing a safe space for reflection and professional development throughout nursing education. Although not explicitly addressed within the framework of the TALK© model, emotions play a vital role, enhancing learning and reflection on real experiences. Debriefing also normalizes emotions, fostering trust and understanding. Managing judgment and guilt is critical, with facilitator neutrality and peer support mitigating these feelings.

Since existing studies on the TALK© tool have primarily been conducted with healthcare professionals and have relied predominantly on quantitative methodologies, there is a clear need to explore nursing students’ perceptions of this structured reflection and learning tool. Gaining insight into how nursing students experience and interpret the TALK© model may provide valuable evidence regarding its educational effectiveness and inform its potential integration into undergraduate nursing curricula.

## Figures and Tables

**Figure 1 nursrep-15-00194-f001:**
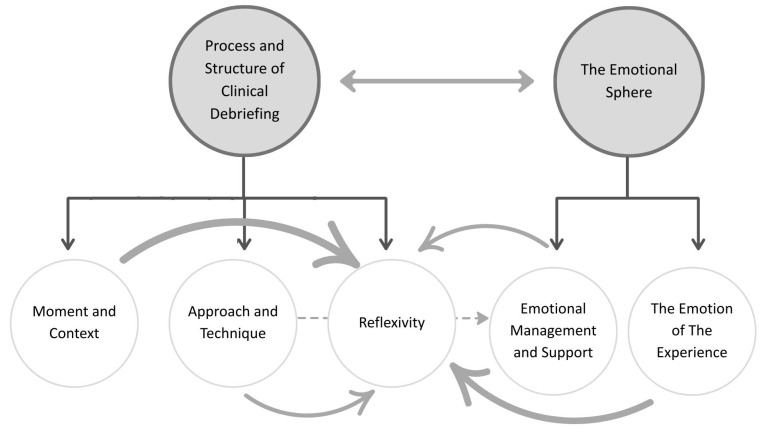
Thematic map of qualitative findings.

**Table 1 nursrep-15-00194-t001:** Semi-structured interview guide.

Topic to Explore	Question
Feelings and emotions	How did you feel during the debriefing? How does it influence your comfort that the debriefing is not of a simulated situation? How did you feel during the debriefing when speaking in front of other people? How do you think feelings and emotions may vary depending on whether or not you are the protagonist of the incidents you discussed?
Relationship established between classmates	How did you feel about your colleagues during the debriefing? How did being with the same colleagues from your clinical internship influence your experience during the debriefing? How was your relationship during the internship and how did it influence the development of the debriefing?
Facilitators’ role	What is your opinion about how the facilitators handled the debriefing? If the facilitators had begun by sharing a challenging professional experience of their own as nurses, how do you think that might have influenced the debriefing process? Did you feel guilty or judgmental at any point, and can you tell me what made you feel/not feel that way? If you did feel judged or guilty, what changes would you suggest in how the debriefing is conducted to reduce or prevent that feeling? How did you experience the fact that the facilitators were people you already knew? How did it affect the debriefing that the facilitators presented themselves as “non-teachers”? What impact did it have on the debriefing that the facilitators were not directly involved in or present during the incidents being discussed? How do you think you would feel doing the debriefing with your assistive tutors?
Time for debriefing	Considering that the debriefing did not take place on the same day as the incidents you mentioned, when do you think is the best time to carry out this type of debriefing and why? What differences do you think there are between doing this debriefing in one service or another?
Perception about the TALK© tool	What is your opinion about the TALK© tool that was used to carry out the debriefing? What differences do you find between this debriefing and the conversations you have with each other during the breaks? What do you think about the fact that you were the ones who chose the topics to talk about in the debriefing? How do you think it would have influenced you if the topics had been imposed by the facilitators? Do you find any differences between the educational debriefing you have done in simulation and the clinical debriefing? How do you think the fact that the debriefing was based on a real situation you experienced—rather than a simulated one—affected your experience?
Impact of debriefing with a good judgment approach on student	How did you experience the questions asked by the facilitators? What was your feeling when the facilitators asked you about your emotions and feelings when talking about the incidents? How did that influence the debriefing?
Issues covered in the clinical debriefing	What problems were you able to detect in the above incidents through the clinical debriefing? Did you talk about all the issues you wanted to talk about?

**Table 2 nursrep-15-00194-t002:** Focus groups, number of participants per focus group, and duration of each focus group.

Focus Group	Participants	Duration (Minutes)
Intensive Care Unit (ICU)	11 participants	130
Operating Room (OR)	6 participants	86
Emergency Room (ER)	10 participants	90

**Table 3 nursrep-15-00194-t003:** Sociodemographic data.

Participant (P)	Sex	Age	Clinical Rotation Unit/Clinical Debriefing Session
P2	Female	26	OR
P5	Female	22	ER and ICU
P6	Male	22	ER
P7	Female	22	ER
P8	Female	22	ER
P9	Female	22	ER
P10	Female	22	ER
P11	Female	22	ER
P12	Female	25	ER
P13	Female	23	ER
P14	Female	22	ER
P15	Female	49	OR
P16	Female	22	OR
P17	Female	36	OR
P18	Female	22	OR
P19	Male	22	OR and ICU
P20	Female	22	OR
P21	Female	22	ICU
P22	Female	22	ICU
P23	Female	22	ICU
P24	Female	22	ICU
P25	Female	23	ICU
P26	Female	22	ICU
P27	Female	22	ICU
P28	Female	22	ICU
P29	Female	22	ICU
P30	Male	22	ICU

## Data Availability

The data from the interviews are not available due to data protection and privacy considerations.

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
