# Peer review of "Nursing Students’ Perceptions of Clinical Debriefing TALK©: A Qualitative Case Study"

_nursrep, 2025, doi:10.3390/nursrep15060194_

Round 1

Reviewer 1 Report

Comments and Suggestions for Authors

Congratulations on the topic under study, training nursing students in patient safety is always a topical issue.
I'll make a few suggestions and observations:
The framework should focus a little on patient safety and the importance of training nurses in this area. As for the technique under study, the framework is well-structured and well-founded.
The methodology is adequately presented and substantiated.
Regarding the development of the study, it is not clear whether the researchers have any academic or internship relationships (with influence over the students) that could influence the students' responses and, therefore, the results. If there is, it should be mentioned in the limitations of the study.
In the results, it would be important to have a table with the themes, sub-themes and examples of excerpts from the participants' verbalisations.
The conclusions are based on the results.
There is a slightly high number of references over 10 years old.

Author Response

Thank you very much for your review of the manuscript. We appreciate your feedback on the work presented and hope to be able to respond appropriately to your requests in the attached document.

Reviewer 2 Report

Comments and Suggestions for Authors

Thank you for the opportunity to review this article. Overall, I find the article well-written and readable, with careful attention to the study's methodological rigor. I appreciated the use of the COREQ standard, which appears to have been properly followed.  My only suggestion would be to compare your study's results with the literature developed over the years on TALK©, in order to contrast and enrich your findings. As per journal rules, I am unable to suggest references, although several are available in the literature. This comparison could provide additional insights and place your work within the broader existing scientific context.

Author Response

(The authors gave the same response as above.)

Reviewer 3 Report

Comments and Suggestions for Authors

Dear authors,

I think you have made a good qualitative approach to a methodology of reflection very appropriate for nursing students. As a proposal for improvement, for the future, it would be to analyse the long-term impact of the TALK© model on the performance or retention of clinical knowledge, for which it would perhaps be convenient to choose third year students, as fourth year students are lost track of.

Author Response

(The authors gave the same response as above.)

Reviewer 4 Report

Comments and Suggestions for Authors

Dear Authors, this manuscript offers valuable insights into how undergraduate nursing students experience clinical debriefing using a specific tool. The study is contextually relevant, methodologically well-structured, and provides an important contribution to the under-researched area. However, several issues should be addressed to improve the manuscript's rigor, clarity, and contribution to the literature:

1) While the use of Braun and Clarke’s thematic analysis framework is appropriate, the process of theme generation remains somewhat vague. Although you state that themes were developed inductively, the coding process and the transition from initial codes to themes are not described in detail. Providing a clearer outline of the analytical steps, including how coding was approached and how consensus was reached among the researchers, would enhance transparency. Including a visual representation of the coding process, such as a thematic map or coding matrix, could further help the reader.

2) The decision to move away from the traditional notion of data saturation is acceptable but would benefit from further clarification. The rationale for this choice, particularly given the qualitative case study approach and the sample size, should be explained more clearly. Readers unfamiliar with newer conceptualizations of saturation might question the completeness of the analysis without a justification.

3) While the research team is clearly well-qualified and multidisciplinary, the roles of the authors during the debriefing process are somewhat unclear. It would be useful to clarify whether any of the researchers conducting the data collection also served as facilitators during the debriefing sessions. This detail is crucial for evaluating the neutrality and potential influence of the research team, especially in a study dealing with emotionally charged experiences and perceptions.

4) Many of the references cited in the introduction and discussion sections are significantly outdated. Foundational texts such as those by Dewey (2011), Gibbs (1988), Johns (2002), and Schön (1983, 1987) are important, but their inclusion would benefit from supplementation with more recent evidence-based studies. Several citations date back over a decade or more, including Berterö (2010), Carrillo Pineda et al. (2011), and Denzin and Lincoln (2012). To align with contemporary academic standards and strengthen the relevance of the discussion, I recommend incorporating recent literature published within the last five years. The reference supporting the Declaration of Helsinki is outdated. Additionally, the manuscript should be formatted in accordance with the MDPI style.

5) The results section tends to be repetitive at times. Some sentences are overly long and offer limited analytical depth. Rather than including extensive narrative excerpts, the manuscript would benefit from a more concise selection of quotes, each followed by deeper interpretative commentary. There are also a few stylistic and linguistic concerns. Although the manuscript is generally well written, several phrases would benefit from revision to improve clarity and academic tone. I suggest a careful proofreading of the manuscript, possibly with the assistance of a native English speaker or professional language editor.

6) The manuscript would be strengthened by adding a brief paragraph at the end of the discussion that addresses the practical implications of the findings for clinical education and policy. The limits should be also enhanced. This would underscore the relevance of the study for stakeholders beyond the academic context and highlight potential avenues for implementation or further research.

Comments on the Quality of English Language

Although the manuscript is generally well written, several phrases would benefit from revision to improve clarity and academic tone. I suggest a careful proofreading of the manuscript, possibly with the assistance of a native English speaker or professional language editor.

Author Response

(The authors gave the same response as above.)

Reviewer 5 Report

Comments and Suggestions for Authors

First and foremost, I would like to express my sincere appreciation to the editors of Nursing Reports (MDPI) for the opportunity to review this manuscript. I firmly believe that peer review constitutes a vital element in assuring the scientific quality of published work, providing an impartial perspective on the manuscript and aiming to support the enhancement of its rigour and alignment with the journal’s and MDPI’s editorial standards.

General Perspective
This manuscript presents a highly relevant piece of evidence for nursing educators, as it fosters deep reflection and critical thinking among nursing students regarding their actions and experiences during a critical stage of their professional development. Moreover, it contributes meaningfully to the promotion of a culture of systematic analysis of critical incidents in educational and clinical settings, including errors, as part of a broader quality improvement framework.

Introduction
The introductory section provides a suitable overview of the topic and introduces the TALK© tool. However, it lacks a clear and explicit statement of the study aim(s) at its conclusion. I recommend the inclusion of a concise sentence outlining the main objective(s) of the study.

Materials and Methods
As in the previous section, the introductory paragraph mentions that the research questions guided the methodological approach. Nonetheless, these research questions are not presented in the manuscript. I suggest that the authors explicitly state them to help the reader understand the focus of the investigation.

Other than this, no further concerns need to be raised regarding the methodological section.

Results
In qualitative studies, the Results section often poses challenges in terms of organisation due to the richness and volume of data. This manuscript, however, is clearly written and logically structured. That said, although the authors state that 24 categories were identified, these are not easily discernible throughout the text. I strongly recommend that the authors include a table (e.g., Table 2) clearly outlining the thematic structure, main themes, subthemes, categories, and their corresponding excerpts or references from participants.

Discussion
The Discussion section meets the expected depth and breadth required for this type of study. However, I would encourage the authors to further elaborate on the study’s limitations. Specifically, what is currently described as a limitation (“focus on a single nursing course and only fourth-year students’ perspectives”) is, in fact, a methodological characteristic rather than a limitation per se. A true limitation concerns aspects that might undermine or constrain the study’s findings or generalisability, given the chosen design and context.

Additionally, I would welcome some reflection on the potential avenues for future research that could stem from this study.

Conclusion
No specific issues noted.

References
The reference list is appropriate and up to date.

Author Response

(The authors gave the same response as above.)

Round 2

Reviewer 2 Report

Comments and Suggestions for Authors

Thank you for the opportunity to review this article. I appreciate the authors' responsiveness in addressing the initial feedback. The revised manuscript has improved significantly, particularly in aligning more closely with the COREQ criteria, and the tables, figures, and references now appear appropriate and well-integrated.

Main Research Question:
The main question addressed by the research is to explore the experiences of fourth-year nursing students during clinical internships, specifically focusing on the use of the TALK© debriefing technique.

Originality and Relevance:
The topic is relevant and timely within the field of nursing education and patient safety. It addresses a specific gap related to the use of structured debriefing tools in undergraduate nursing training. While debriefing techniques have been studied extensively, the application of TALK© in this context appears to be less explored and thus contributes original insights.

Contribution to the Field:
This study offers a valuable contribution by examining the subjective experiences of nursing students using a structured debriefing model. The inductive thematic analysis and use of multiple qualitative data sources enhance the richness of the findings. The inclusion of the TALK© framework within undergraduate education can represent an important pedagogical approach.

Methodological Improvements:
The study demonstrates methodological rigor, particularly in its use of multiple data collection methods and a well-defined analysis strategy. The authors have improved adherence to the COREQ standards in the revised version. 

Conclusions and Alignment with Evidence:
The conclusions are consistent with the evidence presented and appropriately address the main research question. The themes identified—such as the reflexive process, emotional impact, and contextual factors—are clearly supported by participant quotes and analysis.

References:
The references are appropriate, current, and have been improved in the revised version. They support the study’s theoretical and methodological framework well.

Tables and Figures:
The tables and figures are clear, well-structured, and contribute effectively to the presentation of results. They support the text and do not require further revision.